# Pretreatment with P2Y_12_ Receptor Inhibitors in Acute Coronary Syndromes—Is the Current Standpoint of ESC Experts Sufficiently Supported?

**DOI:** 10.3390/jcm12062374

**Published:** 2023-03-19

**Authors:** Piotr Niezgoda, Małgorzata Ostrowska, Piotr Adamski, Robert Gajda, Jacek Kubica

**Affiliations:** 1Department of Cardiology and Internal Medicine, Ludwik Rydygier Collegium Medicum, Nicolaus Copernicus University in Toruń, 85-094 Bydgoszcz, Poland; 2Gajda-Med Medical Center, 06-100 Pułtusk, Poland

**Keywords:** pretreatment, acute coronary syndrome, NSTEMI, STEMI, P2Y_12_ receptor inhibitors

## Abstract

Excessive platelet reactivity plays a pivotal role in the pathogenesis of acute myocardial infarction. Today, the vast majority of patients presenting with acute coronary syndromes qualify for invasive treatment strategy and thus require fast and efficient platelet inhibition. Since 2008, in cases of ST-elevation myocardial infarction, the European Society of Cardiology guidelines have recommended pretreatment with a P2Y_12_ inhibitor. This approach has become the standard of care in the majority of centers worldwide. Nevertheless, the latest guidelines for the management of patients presenting with acute coronary syndrome without persisting ST-elevation preclude routine pretreatment with the P2Y_12_ receptor inhibitor. Those who oppose pretreatment support their stance with trials failing to prove the benefits of this strategy at the cost of an increased risk of major bleeding, especially in individuals inappropriately diagnosed with an acute coronary syndrome, thus having no indication for platelet inhibition. However, adequate platelet inhibition requires even up to several hours after application of a loading dose of P2Y_12_ receptor inhibitors. Omission of data from pharmacokinetic and pharmacodynamic studies in the absence of data from clinical studies makes generalization of the pretreatment recommendations difficult to accept. We aimed to review the scientific evidence supporting the current recommendations regarding pretreatment with P2Y_12_ inhibitors.

## 1. Introduction

Excessive platelet activation plays a major role in the pathogenesis of acute myocardial infarction (AMI) either with ST elevation (STEMI) or non-ST elevation (NSTEMI) [1]. STEMI is typically associated with a sudden, total occlusion of a coronary artery with a thrombus which forms on a ruptured atherosclerotic plaque [2]. Destruction of the natural integrity of coronary endothelium exposes lipids, building plaque as well as the collagen fibers which are responsible for the initiation of platelet activation leading to the formation of a clot. Contrary to STEMI patients, those presenting with NSTEMI are generally expected to have critical stenosis of a particular coronary artery rather than total occlusion [3].

The latest European Society of Cardiology (ESC) guidelines on the management of patients presenting with both STEMI and NSTEMI recommend a 12-month course of dual antiplatelet therapy (DAPT) comprising aspirin and a P2Y_12_ receptor inhibitor [4,5]. Among the currently available P2Y_12_ receptor inhibitors, two thienopyridines—clopidogrel and prasugrel—are inactive pro-drugs that require hepatic activation through more (clopidogrel) or less complex (prasugrel) metabolic pathways, whereas ticagrelor and cangrelor are active and reversible agents. Cangrelor is the only intravenous P2Y_12_ inhibitor characterized by the rapid onset and offset of action [6].

The term “pretreatment” with a P2Y_12_ receptor inhibitor is used to describe the strategy of the loading dose administration at first medical contact, after the diagnosis of acute coronary syndrome (ACS) is made without any data on the coronary anatomy [7]. Such an approach was recommended in a previous edition of the non-ST elevation (NSTE) ACS 2015 ESC guidelines irrespective of the initial therapeutic strategy, with the exception of prasugrel, which was limited to individuals who qualified for PCI after coronary angiography [8]. The potential benefits of pretreatment with a P2Y_12_ inhibitor include a reduction in the rate of ischemic events while waiting for invasive treatment, the prevention of early-stent thrombosis, and a reduction in glycoprotein IIb/IIIa bail-out use. [9] On the other hand, patients preloaded with P2Y_12_ inhibitors are considered to be burdened with an increased risk of bleeding, especially if femoral access is used for coronary angiography or if further cardiac surgery is required. Therefore, the optimal timing of P2Y_12_ receptor inhibitor administration has become a subject of scientific debate. Early methods of platelet inhibition included the administration of Glycoprotein IIb/IIIa inhibitors at different time-points after the diagnosis of ACS. This standpoint was based on findings from several trials in which particular agents’ use was associated with better clinical outcomes. There was, however, an increased risk of bleeding events when Gp IIb/IIIa were administered [9,10,11]. Based on the latest issues of ESC guidelines for both STEMI and NSTE-ACS, prehospital administration of these agents is not recommended due to lack of benefit and increased bleeding rates [4,5]. The first trial to provide beneficial effects of pretreatment with clopidogrel in NSTE-ACS patients was the PCI-CURE study. Pretreatment resulted in lower rates of death, myocardial infarction, refractory ischemia, and urgent revascularization. There was no significant increase in bleeding episodes in the clopidogrel arm in comparison with the placebo arm. In this study, clopidogrel was administered very early, before PCI, which contributed to beneficial effects of pretreatment, taking into account that this agent requires up to 24 h to inhibit platelet function if a 300 mg dose is administered or 5–7 days in case of a 75 mg dose [12]. Moreover, the latest ESC guidelines no longer recommend upstream therapy with these agents in NSTE-ACS patients [4], considering it that novel P2Y_12_ receptor inhibitors require only 1–2 h to sufficiently inhibit platelet activity in the majority of patients. However, it has been shown that in case of STEMI, concomitant therapy with opioids, in critically ill patients, or in those presenting with cardiogenic shock, effective platelet inhibition with prasugrel or ticagrelor may require more than 4 h or even may not be achieved within the first 24 h in the most severe cases [13,14]. A schematic presentation of the arguments for and against pretreatment is shown in Figure 1.

## 2. Pretreatment in STEMI

According to the ESC Guidelines, patients diagnosed with STEMI should qualify for primary PCI. This strategy assumes immediate access to 24/7 hemodynamic facilities with availability of trained and adequately equipped ambulance teams to diagnose STEMI, administer initial pharmacotherapy, and stabilize a patient if necessary [15,16,17]. A strong recommendation to perform primary PCI is valid for patients with a recent onset of symptoms, i.e., less than 12 h if there is a persistent ST segment elevation or even over 12 h in cases of ongoing symptoms of ischemia, life-threatening arrhythmias, or hemodynamic instability—**class of recommendation I, level of evidence A** [18,19]. Moreover, primary PCI should be considered in STEMI patients even if they present with typical symptoms up to 48 hours—**class of recommendation IIa, level of evidence B** [20,21]—while ultimately, only asymptomatic cases with a late diagnosis, i.e., over 48 h after AMI, should be disqualified from primary PCI—**class of recommendation III, level of evidence A** [22,23]. As stated in the ESC guidelines, prasugrel or ticagrelor should be administered before (or at least at the time of) PCI in STEMI patients. In the case of unavailability of those two agents, clopidogrel should be used [5]. Nevertheless, data supporting such a standpoint are limited. The recommendation is based on the fact that pretreatment with either prasugrel or ticagrelor was allowed in studies, which led to the approval of those agents with the TRITON-TIMI 38 trial and the PLATO trial, respectively [24,25]. The trials referred to in the ESC guidelines are marked with (*).

The CIPAMI trial (*), a small clinical study conducted by Uwe et al. and published in 2012, aimed to evaluate the clinical effects of pretreatment with clopidogrel in STEMI patients. Overall, 337 subjects were enrolled and randomized to receive a loading dose of clopidogrel in the prehospital phase (*n* = 166) or after coronary angiography, directly prior to PCI (*n* = 171). The study revealed no significant differences in terms of primary endpoint, which was defined as TIMI 2/3 patency in the culprit vessel before PCI (49.3% in the pretreatment arm vs. 45.1% in the no-pretreatment arm, *p* = 0.5). Moreover, rates of TIMI 3 in a culprit vessel before PCI did not differ significantly between study arms (32.6% vs. 27.4%, respectively, *p* = 0.3). Additionally, the difference in the rate of a composite of death, re-infarction, and urgent target vessel revascularization was not statistically significant (3.0% vs. 7.0%, *p* = 0.09); however, a trend favoring pretreatment was clearly visible. It must be underlined that no increase in major bleeding complications in the pretreatment arm was found (9.1 vs. 8.2%, *p* = 0.8) [26].

A multicenter Austrian registry (*) of patients undergoing primary PCI due to STEMI [27] evaluated the clinical outcomes of pretreatment with clopidogrel. A total of 5955 patients were included in the analysis based on clopidogrel administration strategy, pretreatment (*n* = 1635), or periprocedural use (*n* = 4320). Pretreated individuals had a lower rate of in-hospital mortality (*p* < 0.01) when compared to the no-pretreatment arm. Moreover, the risk of bleeding was not significantly increased (*p* = 0.90).

A sub-analysis of STEMI patients undergoing primary PCI who were identified in the Swedish Coronary Angiography and Angioplasty Registry (SCAAR) (*) was performed to evaluate the effects of pretreatment with clopidogrel on the reduction in the 1-year death/MI rate. Overall, 13,847 patients were included in the analysis. The rates of 1-year death/MI as well as 1-year death alone were significantly reduced (HR 0.82, 95% CI 0.73–0.93 and HR 0.76, 95% CI: 0.64–0.90, respectively); however, no reduction was observed in 1-year MI (HR 0.90, 95% CI 0.77–1.06). Data regarding bleeding were available in 12,548 patients. The risk of bleeding was similar in the analyzed arms [28].

Another study, the Load&Go randomized trial conducted by Ducci et al., tested the clinical efficacy of the prehospital administration of two doses of clopidogrel, 600 mg or 900 mg, vs. the periprocedural use of 300 mg of clopidogrel in STEMI patients undergoing primary PCI. The study population included 168 participants randomized in a 1:1:1 ratio to receive (1) no pretreatment, (2) 600 mg of clopidogrel in the prehospital phase or (3) 900 mg of clopidogrel in the prehospital phase. The study failed to prove the benefits of pretreatment in STEMI patients. The rate of primary endpoint, thrombolysis in myocardial infarction perfusion grade 3 (TMPG 3), did not differ significantly (64.9% for pretreatment with either 600 mg or 900 mg in the clopidogrel arm vs. 66.1% in the no-pretreatment arm; *p* = 0.88), and there were also no significant differences between 600 mg vs. 900 mg of clopidogrel in terms of TMPG 3 rate (57.1% vs. 72.7%, respectively; *p* = 0.12). The results of the study also did not reveal any significant differences between rates of bleeding episodes. Platelet reactivity (in platelet reactivity units—PRUs) assessed with the Verify-Now tool was comparable between the pretreatment vs. no-pretreatment arms (342 ± 59 in pretreated individuals vs. 333 ± 72 in the no-pretreatment arm; *p* = 0.20). A direct comparison between the 900 mg, 600 mg, and no pretreatment groups also revealed no differences (337 ± 48 vs. 356 ± 52 vs. 333 ± 72; *p* = 0.080, respectively) [29].

The only randomized trial aiming to evaluate the outcomes of ticagrelor administration at different timepoints in STEMI patients was “The Administration of Ticagrelor in the Cath Lab or in the Ambulance for New ST Elevation Myocardial Infarction to Open the Coronary Artery”—the ATLANTIC trial (*) [30]. Overall, 1862 patients with a recent diagnosis of STEMI (<6 h) were randomized to receive a loading dose of ticagrelor either during transport to the cath lab (prehospital) or directly prior to coronary angiography in the cath lab (in-hospital). The study showed no significant differences in either of the co-primary endpoints, with an absence of at least 70% ST segment resolution before PCI was observed in 86.8% and 87.6% of patients in the prehospital and in-hospital arms, respectively (*p* = 0.63), and an absence of the thrombolysis in myocardial infarction (TIMI) 3 flow at initial angiography was found in 82.6% and 83.1% of patients, respectively (*p* = 0.82). Among secondary endpoints, a very pronounced trend favoring prehospital administration of ticagrelor was observed in terms of number of patients who did not achieve at least 70% ST segment resolution after PCI—42.5% vs. 47.5% in the prehospital and in-hospital arms, respectively, *p* = 0.05. The bleeding rates were nearly identical between the study arms, while definite stent thrombosis occurred significantly more often in the in-hospital arm (0 vs. 8 patients, *p* = 0.008, within 24 h post PCI and 2 vs. 11 patients, *p* = 0.02, within 30 days post PCI in the prehospital and in-hospital arms, respectively) [30].

The results of a PCI-CLARITY randomized trial conducted by Sabatine et al. showed benefits of the early administration of clopidogrel in STEMI patients on fibrinolytic therapy. Patients underwent randomization in a 1:1 ratio into two study arms: (1) Administration of a loading dose of 300 mg of clopidogrel followed by 75 mg daily or (2) administration of a placebo. Treatment was continued until coronary angiography, i.e., 2–8 days after the index event. Pretreatment with clopidogrel was associated with a lower rate of MI or stroke before PCI than it was in the placebo arm (4.0% vs. 6.2%, respectively, *p* = 0.03). The difference between study arms was also significant with regard to a composite of cardiovascular death, MI, or stroke after PCI (3.6% vs. 6.2%, respectively, *p* = 0.008). Overall, the rate of a composite of cardiovascular death, MI, or stroke before and after PCI was significantly lower in the pretreatment arm than in the placebo group (7.5% vs. 12%, respectively, *p* = 0.001). Throughout the study, rates of both major and minor TIMI bleeding episodes did not differ significantly between pretreatment and placebo arms (0.5% vs. 1.1%, *p* = 0.21, and 1.4% vs. 0.8%, *p* = 0.26, for major and minor TIMI bleeding, respectively) [31].

In a recently conducted multicenter randomized ISAR-REACT 5 trial, the first head-to-head comparison of ticagrelor and prasugrel, a total of 4018 patients with a diagnosis of ACS were randomized to receive a predefined P2Y_12_ receptor inhibitor in a 1:1 ratio [32]. The study did not directly compare pretreatment and delayed loading with antiplatelet agents. The major assumption in terms of pretreatment was based on the fact that patients treated with prasugrel were loaded with the drug after diagnostic coronary angiography, while those in the ticagrelor arm generally received pretreatment. Overall, the study population included 41.1% STEMI patients. However, the primary endpoint of the trial—1-year incidence of a composite of death, MI, or stroke—although numerically higher for ticagrelor, was not statistically significant (*n* = 83 (10.1%) vs. *n* = 64 (7.9%); odds ratio (OR) 1.31 [0.94–1.81], *p* = ns). It must be highlighted, however, that the ISAR-REACT 5 study caused multiple controversies and became the subject of vivid scientific discussions mainly due to serious limitations, including improbably high adherence to treatment, controversial follow-up of the patients (only 10% underwent in-center visits), and an unacceptably high proportion of participants being excluded from certain steps of the analysis [33,34].

## 3. Pretreatment in NSTE-ACS

The latest issue of the ESC Guidelines for the management of NSTE-ACS patients brought about a major change in terms of pretreatment with P2Y_12_ receptor inhibitors. With the publication of the document, routine pretreatment became not recommended (**class of recommendation III, level of evidence A**) [4]. Its authors, however, note that despite the unquestionable necessity to achieve early and efficient platelet inhibition in NSTE-ACS patients undergoing PCI, it is mainly due to the lack of large clinical trials supporting pretreatment that all practitioners should change their habits from now on. Data on pretreatment can be obtained from five randomized controlled trials, one registry, and three meta-analyses, among which only three are referred to in the latest issue of the ESC Guidelines—these studies were marked with (*).

The study that is being referred to by opposers of pretreatment is the abovementioned ISAR-REACT 5 trial (*). Among all the participants, 42.6% presented with NSTEMI, while 12.7% presented with unstable angina (UA). Taking into account the entire population, the primary endpoint of the study—a composite of death from any cause, MI, or stroke at 1 year after randomization—occurred significantly more often in the ticagrelor arm when compared with the prasugrel arm (9.3% vs. 6.9%, respectively, *p* = 0.006). As far as treatment safety is concerned, rates of Bleeding Academic Research Consortium (BARC) type 3, 4, or 5 bleeding episodes did not differ significantly (5.4% vs. 4.8% for ticagrelor and prasugrel, respectively, *p* = 0.46). According to the authors, the factor that mainly contributed to the final results was the difference in number of MI events, which was noticeably lower in the prasugrel arm than in the ticagrelor arm (*n* = 60 (3.0%) vs. *n* = 96 (4.8%), respectively, hazard ratio (HR) 1.63 [1.18–2.25]). To summarize, the presented results of the ISAR-REACT 5 trial do not promote pretreatment. Except for the previously mentioned limitations that bias the construction of the study and data analysis, no definite analysis of pretreatment vs. in-hospital administration of P2Y_12_ was performed, but it was rather a consequence of the fact that patients treated with prasugrel were loaded with the drug only after diagnostic angiography and qualification for PCI, while patients on ticagrelor were allowed to receive it earlier [32].

The Comparison of Prasugrel at the Time of Percutaneous Coronary Intervention or as Pretreatment at the Time of Diagnosis in Patients with Non-ST Elevation Myocardial Infarction (ACCOAST) trial (*) was a large clinical study conducted by Montalescot et al. aiming to evaluate the effects of administration of prasugrel immediately after making the diagnosis of NSTE-ACS or just after diagnostic coronary angiography in patients who qualified for PCI. A total of 4033 patients were enrolled in the study. They had to be diagnosed with NSTEMI and qualify for invasive angiography 2–48 h after randomization, which was performed in a 1:1 ratio to the following two groups: (1) The pretreatment group, in which patients received 30 mg of prasugrel before angiography and another 30 mg in case of indication for PCI and (2) the control group, in which patients were given a placebo before coronary angiography and 60 mg of prasugrel if PCI was indicated. If coronary artery bypass graft (CABG) surgery was indicated, individuals in the pretreatment group were not given the additional 30 mg of prasugrel, and those in the control group did not receive prasugrel at all. Safety outcomes included TIMI bleeding episodes, which were analyzed to determine whether they were related to CABG or not. There were no differences between the study arms in terms of a composite of cardiovascular death, MI, stroke, urgent revascularization, or rescue use of glycoprotein IIb/IIIa within 7 days following randomization (10.0% vs. 9.8%, *p* = 0.81 for the prasugrel and control groups, respectively). Moreover, rates of particular components of the primary endpoint did not differ significantly either at 7 days or at 30 days following randomization. Ischemic complications within the period of waiting for coronary angiography occurred in 0.8% of patients in the pretreatment arm and in 0.9% of those in the control arm (*p* = 0.93). In patients who underwent PCI, rates of the primary endpoint also did not differ at 7 or at 30 days post randomization (13.1% vs. 13.1%, *p* = 0.93, at day 7; 14.1% vs. 13.8% at day 30 for the pretreatment and control arms, respectively, *p* = 0.77). Evaluation of rates of bleeding episodes revealed a significant increase in the pretreatment group both at 7 and 30 days after randomization when compared to the control group. All CABG-related and non-CABG-related major TIMI bleeding events occurred in 2.6% vs. 1.4% patients, respectively, *p* = 0.006, at day 7 and in 2.8% vs. 1.5% patients, respectively, *p* = 0.002, at day 30. Significant differences were also observed in non-CABG-related major TIMI bleeding—1.3% vs. 0.5%, respectively, *p* = 0.003, at day 7 and 1.6% vs. 0.6%, respectively, *p* = 0.002, at day 30. Nevertheless, there were groups associated with a lower risk of bleeding throughout the study, especially younger patients (<75 years of age), patients with a body weight over 60 kg, or those who underwent PCI through radial access. The subgroup analysis revealed that in patients who received a loading dose of prasugrel earlier than the median delay time of 15 h post symptom onset, the incidence of primary endpoint was reduced by 24% (0.76, 0.57–1.01, *p* = 0.004) without any significant increase in bleeding episodes (*p* = 0.23). In summary, it must be pointed out that the results of the ACCOAST trial do not support routine pretreatment with prasugrel in NSTE-ACS patients, but they support the idea of pretreatment in individuals with a recent diagnosis of ACS [35].

The aforementioned SCAAR registry (*) [36] is another dataset used to determine the outcomes of pretreatment in NSTE-ACS patients with all available oral P2Y_12_ receptor inhibitors. A total of 64,857 NSTE-ACS patients who underwent PCI procedures were included in the analysis. A total of 59,894 patients (92.4%) were pretreated with a particular P2Y_12_ receptor inhibitor: 43.7% with clopidogrel, 54.5% with ticagrelor, and 1.8% with prasugrel. The primary endpoint of the study was 30-day mortality rate. Data were obtained from the Swedish National Population Registry, which impacts the completeness and reliability of death numbers. There is, however, no detailed information regarding causes of death; thus, only all-cause mortality could be evaluated. Baseline characteristics of the study population revealed a noticeable imbalance between the pretreatment and control arms regarding age, diabetes, arterial hypertension, history of smoking, prior CABG, or history of MI. The analysis of procedural aspects of PCI also revealed non-negligible differences. The percentage of patients undergoing PCI through radial access was significantly lower in the pretreatment arm (78.6% vs. 81.8%, *p* < 0.001) and were more frequently administered GP IIb/IIIa inhibitors (2.6% vs. 1.9%, *p* = 0.002), bivalirudin (15.9% vs. 8.8%, *p* < 0.001), and clopidogrel as the pretreatment agents (45.3% vs. 18.9% for clopidogrel, 52.9% vs. 78.8% for ticagrelor, and 1.8% vs. 2.3% for prasugrel, *p* < 0.001). The primary endpoint of the study, if adjusted only for age and sex, was significantly lower in the pretreatment arm than in the control group (1.4% vs. 2.5%, respectively, *p* < 0.001). After inclusion of the remaining variables into the instrumental variable analysis, i.e., diabetes, prior MI, prior PCI or CABG, smoking status, severity of coronary artery disease, hypertension, hyperlipidemia, indication for PCI, type of P2Y_12_ receptor antagonist, and completeness of revascularization, mortality rates did not differ significantly (adjusted OR 1.44; 95% confidence interval (CI), 0.78–2.62; *p* = 0.36). Similarly, there were no significant differences between study groups in terms of 1-year mortality (4.3% vs. 7.1% for the pretreatment vs. control group, adjusted OR 1.34 (0.77–2.34), *p* = 0.3) or definite stent thrombosis at day 30 (0.2% vs. 0.2%, adj. OR 1.17 (0.64–2.16), *p* = 0.6). Bleeding episodes occurred less frequently in the pretreatment arm than in controls (6.0% vs. 7.5%, respectively), but after the adjustment, the risk was higher in pretreated individuals (adj. OR 1.49 (1.06–2.12), *p* = 0.02). This result remained valid even after exclusion of minor bleeding events (adj. OR 2.31 (1.34–3.98), *p* = 0.002). Moreover, in-hospital bleeding was associated with an increase in 30-day and 1-year mortality rates (adj. OR 8.68 (7.54–9.98), *p* < 0.001 and adj. OR 3.05 (2.73–3.42), *p* < 0.001, respectively).

Despite presenting the real-life data of NSTE-ACS patients in Sweden, the SCAAR registry is biased due to several aspects, including the lack of information regarding patients mistakenly diagnosed with NSTE-ACS, patients who died before admission to the hospital, or patients who were treated with any P2Y_12_ receptor antagonist beforehand. Moreover, the registry lacks subgroup analysis in terms of time since symptom onset or patients’ clinical condition, which impacts urgency for intervention. It also must be pointed out that propensity score matching resulted in a noticeable change in the raw data analysis for both efficacy and safety outcomes.

In a randomized Early and Sustained Dual Oral Antiplatelet Therapy Following Percutaneous Coronary Intervention (CREDO) trial, the authors evaluated outcomes of 12-month clopidogrel use in patients presenting with NSTE-ACS who underwent PCI as well as the potential benefits of pretreatment with this agent. Patients were randomized to receive either 300 mg clopidogrel (*n* = 1053) or placebo (*n* = 1063) between 3 and 24 h preceding coronary angiography. After PCI, all patients received clopidogrel until day 28. Patients who were enrolled in the clopidogrel arm continued therapy with clopidogrel for up to 1 year, while those in the placebo arm were switched to placebo on day 29. The primary endpoint in the CREDO trial was a composite of 1-year mortality, MI, or stroke. A 12-month treatment with clopidogrel reduced the rate of the primary endpoint when compared to the placebo group (8.5% vs. 11.5%, respectively, relative risk reduction (RRR) 26.9% (3.9–44.4), *p* = 0.02). Pretreatment with clopidogrel reduced the risk of the combined endpoint (death, MI, and urgent revascularization of the target vessel) at day 28 by 18.5%, but the difference was not significant (*p* = 0.23). If clopidogrel was administered over 6 h before PCI, the reduction in the combined endpoint was far more pronounced (38.6%, 95% CI; −1.6% to 62.9%, *p* = 0.051), as should be expected taking into account the pharmacokinetics of clopidogrel [37].

Another study, Downstream Versus Upstream Strategy for the Administration of P2Y_12_ Receptor Blockers In Non-ST Elevated Acute Coronary Syndromes With Initial Invasive Indication (DUBIUS), was designed to evaluate differences between upstream (pretreatment) and downstream (no-pretreatment) administration of the potent agents ticagrelor and prasugrel. A total of 1449 patients were randomized in a 1:1 ratio to upstream (with ticagrelor) or downstream therapy. Those in the downstream group who qualified for PCI underwent another randomization to receive prasugrel or ticagrelor. The primary endpoint of the study was defined as a composite of death from vascular causes, nonfatal MI, nonfatal stroke, and major fatal bleeding (BARC type 3, 4, and 5) at day 30 following randomization. There was no significant reduction in the primary outcome of the study between the downstream and upstream groups at 30 days (2.9% vs. 3.3%, respectively, absolute risk reduction (ARR): –0.46; 95% CI: –2.87 to 1.89, *p* = 0.5). BARC 3, 4, and 5 episodes occurred with a similar frequency in both groups. The study was prematurely terminated due to low incidence of events, both ischemic and bleeding. Therefore, as the authors stated in the manuscript, there is a very low likelihood that either of the tested strategies would surpass the other [38].

The authors of the ESC Guidelines stated that patients diagnosed with NSTE-ACS planned for an early invasive strategy (coronary angiography in less than 24 hours) should not be routinely pretreated with a P2Y_12_ receptor inhibitor. This standpoint is supported in the paragraph regarding differential diagnosis of NSTE-ACS, where several serious medical conditions mimicking ACS are listed (Table 1). Increased risk of bleeding in cases of aortic dissection, tension pneumothorax, chest/cardiac trauma, cholecystitis, etc., is undoubtedly an undesired phenomenon. On the other hand, the authors conclude that pretreatment may be considered to pretreat patients without a high bleeding risk who are not planned for an early invasive strategy **(class of recommendation IIb, level of evidence C)**.

The authors of the ESC Guidelines do not mention the three available meta-analyses that investigate the issue of pretreatment in ACS patients. One study by Bellemain-Appaix et al. published in 2014 included seven clinical studies, four randomized controlled trials, three observational studies, and one observational analysis based on data from a randomized controlled trial. A total of 32,383 patients were included. The obligatory criterion for the study to be included in the analysis was that it reported all-cause mortality and major bleeding episodes as outcomes. Of all the included patients, 55% were treated with PCI. Pretreatment with thienopyridines was associated with a non-significant reduction in all-cause mortality (OR 0.90, 95% CI: 0.95–1.07, *p* = 0.24). The difference was more pronounced, but still not significant, in randomized controlled trials (OR 0.78, 95% CI: 0.71–1.14, *p* = 0.39). All patients who were pretreated with thienopyridine had an increased risk of major bleeding by 30–45% (OR 1.32, 95% CI: 1.16–1.49, *p* < 0.0001). This meta-analysis does not support routine pretreatment in NSTE-ACS due to its negative influence on the risk of bleeding without definite benefits in the rate of cardiovascular events [39].

Another meta-analysis by Bellemain-Appaix et al. published in 2012 evaluated the differences in clinical outcomes between patients undergoing PCI due to ACS or chronic coronary syndrome depending on pretreatment with clopidogrel. Overall, 15 studies (6 randomized controlled trials, 2 observational analyses from randomized controlled studies, and 7 observational trials) including 37,814 patients were analyzed. Pretreatment with clopidogrel was not associated with a reduced risk of death when compared to no pretreatment (1.54% vs. 1.97%, respectively; OR, 0.80; 95% CI, 0.57–1.11; *p* = 0.17). Nevertheless, a lower incidence of cardiovascular episodes was observed in pretreated patients (9.83% vs. 12.35%, respectively; OR, 0.77; 95% CI, 0.66–0.89, *p* < 0.001). Rates of major bleeding episodes were not significantly increased in pretreated patients when compared to non-pretreated ones (3.57% vs. 3.08%, respectively; OR, 1.18; 95% CI, 0.93–1.50; *p* = 0.18) [40].

A more recent meta-analysis by Nairooz et al. published in 2017 included 16 trials and 61,517 patients diagnosed with ACS (both STEMI and NSTE-ACS). The aim of the analysis was to compare effects of pretreatment with clopidogrel in individuals treated invasively. At 30 days, the rate of major adverse cardiovascular events was significantly lower in pretreated patients than in those who did not receive pretreatment (7.67% vs. 9.46%, respectively, *p* < 0.0001). Similarly, all-cause mortality was significantly reduced (2.8% vs. 4.1%, *p* = 0.0003). There was no difference in the rate of major bleeding events between study arms (1% vs. 2.78%, *p* = 0.89) [41].

It needs to be mentioned that in terms of pretreatment with oral P2Y_12_ receptor inhibitors, the ESC Guidelines do not refer to any pharmacodynamic studies presenting the delay of adequate platelet inhibition after the administration of a loading dose of a particular agent. Early inhibition of platelet function may be expected in stable patients who receive prasugrel or ticagrelor [42,43]. The IMPRESSION trial revealed that even 4 h after the administration of a loading dose of ticagrelor followed by morphine, the percentage of high-platelet-reactivity patients can reach unpredictably high levels (20%, 37%, and 23% for multiple electrode aggregometry, VASP, and Verify-Now, respectively). Even if only patients who did not receive any morphine were taken into account, those numbers reached 17%, 17%, and 8%, respectively [13]. Similar worrisome results were found by Schoergenhofer et al. in a trial that tested the pharmacodynamics of prasugrel in critically ill patients admitted to the intensive care unit. Among them, poor response to prasugrel resulting in a high percentage of individuals with platelet reactivity was very common (65%, 95% CI, 43–84%). Moreover, low plasma concentrations of both prasugrel and its active metabolite were found among study participants. As was found in the study, high plasma concentrations of c-reactive protein were associated with a lower peak plasma concentration of prasugrel (r = −0.51, *p* = 0.02) [14].

A brief summary of the studies described in the text is presented in Table 2.

## 4. Discussion

Undoubtedly, contemporary scientific data regarding pretreatment in ACS are scarce. There are multiple aspects to consider when making the right decision resembles walking on a thin line balancing between increased risk of bleeding and greater ischemic complications. Pretreatment with a P2Y_12_ receptor inhibitor may subconsciously seem to be an obvious approach due to its reasonable rationale. Patients with a new diagnosis of STEMI will most commonly require percutaneous treatment as the majority of cases are caused by total occlusion of a coronary artery. Early inhibition of platelet function plays a pivotal role in this setting.

The main factor that negatively influenced the results of the ALTANTIC trial was a short time difference (31 min) between the tested therapeutic strategies [30]. As mentioned above, registered clinical trials for both prasugrel and ticagrelor allowed pretreatment, which, consistently with pharmacokinetic/pharmacodynamic studies, supports the early administration of P2Y_12_ receptor inhibitors in STEMI.

Contrary to STEMI, the case of NSTE-ACS patients receiving pretreatment with a P2Y_12_ receptor inhibitor has recently become a subject of numerous debates. Unfortunately, superficial analysis of the available data from various clinical studies may lead to misleading assumptions. The authors of the latest issue of the ESC Guidelines for the management of patients presenting with NSTE-ACS no longer recommend routine pretreatment with P2Y_12_ receptor inhibitor based on the ISAR-REACT 5 study results. However, the authors did not take into account several critical limitations regarding this study. Despite being an international, multicenter study, ISAR-REACT 5 was conducted only in two countries with an unacceptable disproportion in the distribution of study sites (21 sites in Germany and only 2 in Italy). Moreover, adherence to treatment exceeded 99%, which makes it hardly believable (in the PLATO trial, which was a registration trial for ticagrelor, the adherence was 82.8%). Controversies in terms of the design of the study are also associated with the schedule of follow-up visits. Only 10% attended an on-site visit, while the following 83% were contacted by telephone and the remaining 7% by mail. Moreover, due to the fact that the analysis of the results was based on an intention-to-treat method, the results were undoubtedly impacted by the fact that over 20% of participants were discharged from the hospital with a different treatment agent than they were assigned to at randomization. As it turned out later, the intention-to-treat method led to the inclusion of 1299 patients who were not treated with the medication they were initially assigned to into the analysis. Taking into account the above, as well as the fact of exclusion of unacceptably high numbers of participants from the final analysis, it is difficult to call the ISAR-REACT 5 trial results ground-breaking [34].

There is common approval for the results of the ACCOAST trial. The strategy of limiting the administration of prasugrel only to patients who are candidates for PCI after diagnostic coronary angiography was the standard in both the 2015 and 2020 ESC Guidelines for the management of patients presenting with NSTE-ACS. Pretreatment with prasugrel was not associated with the reduction in the primary efficacy endpoint of the study but was associated with an increased risk of bleeding. Nevertheless, these results remain valid only if no subgroup analysis is taken into account. The ACCOAST trial clearly does not support routine pretreatment with prasugrel in NSTE-ACS, but noticeable improvement in clinical outcomes is seen in patients who received the loading dose of this agent early after symptom onset [4,8].

With regard to the SCAAR registry, which is a valuable source of data regarding pretreatment and potential benefits as a result of it, bleeding episodes included all events such as: cardiac tamponade, prolonged compression treatment, surgical intervention, a decrease in hemoglobin of at least 2 g/dL, pseudoaneurysms, puncture site hematomas, or transfusions, all classified as BARC type 2 or 3. Despite being consistent with other Swedish registries, such classification impacts the statistics mainly due to the increase in the rate of minor episodes. Moreover, there was a significantly higher percentage of patients in the pretreatment arm who underwent procedures through other than radial access and thus were more predisposed to bleeding complications. It is also worth noting that except for increased risk of bleeding, pretreatment was not worse in terms of efficacy endpoints.

It may be assumed that in cases of such a clear standpoint precluding routine upstream administration of oral P2Y_12_ receptor inhibitors, the authors of the latest issue of the ESC Guidelines for the management of patients presenting with NSTE-ACS would strongly support cangrelor as a solution to numerous aspects of potentially inefficient antiplatelet therapy. Administration of this intravenous inhibitor was associated with a reduction in ischemic events, including stent thrombosis [44,45]. Due to its rapid onset and offset of action, cangrelor has the potential to solve all the aforementioned issues regarding pretreatment. Nevertheless, as stated in the document, administration of cangrelor may be considered in P2Y_12_-naïve patients undergoing PCI (**class of recommendation IIa, level of evidence A**). Another promising approach to achieve quick and reversible platelet inhibition may be a subcutaneous administration of a novel P2Y_12_ receptor inhibitor, selatogrel. In phase 1 and phase 2 studies, this agent successfully inhibited platelet activity in approximately 90% of patients as fast as 30 min after self-administration. Subcutaneous administration of the drug potentially allows to overcome all previously described limitations of oral agents. Nevertheless, to date, the drug has not been approved by either the Food and Drug Administration (FDA) or the European Medicines Agency (EMA) [46,47,48,49]. However, most probably, the wide availability of these agents for in-hospital use would result in the termination of the dispute regarding pretreatment in NSTE-ACS patients [4].

## 5. Summary

As stated in the ESC Guidelines, “although a rationale for pretreatment in NSTE-ACS may seem obvious, for achieving sufficient platelet inhibition at the time of PCI, large-scale randomized trials supporting a routine pretreatment strategy with either clopidogrel or the potent P2Y_12_ receptor inhibitors—prasugrel and ticagrelor—are lacking”. Nevertheless, the authors conclude that “Based upon the available evidence, it is not recommended to administer routine pretreatment with a P2Y_12_ receptor inhibitor in NSTE-ACS patients in whom coronary anatomy is not known and an early invasive management is planned”. Successful treatment of ACS patients is definitely a complex issue comprising multiple multi-directional aspects. The issue of pretreatment with a P2Y_12_ receptor inhibitor was the subject of numerous randomized controlled or observational clinical trials, as well as observational registries. The aim of this review was to discuss the latest recommendations included in the ESC guidelines based on the available data obtained from particular clinical studies. It is hard to agree with the standpoint presented in the guidelines as a generalization for all NSTE-ACS patients as the arguments supporting it seem far too weak. Several questions remain unanswered after analysis of the available data on pretreatment:-Who may benefit from pretreatment and for whom would this strategy be harmful?

Based on the results of the ACCOAST trial, patients receiving early pretreatment with P2Y_12_ receptor inhibitor are expected to benefit most from such strategy regardless of the type of ACS. On the other hand, pretreatment administered late after the onset of symptoms may be harmful. Therefore, it is not recommended.


-Which approach is the most appropriate in the highest-risk patients?


Highest-risk patients are often characterized with an impaired absorption from the gastrointestinal tract due to multiple causes including centralization of circulation or concomitant therapy with opioids, which makes parenteral administration of P2Y_12_ receptor inhibitors the best approach.

As a simple and generalized point of view may be misleading due to diversity of the NSTE-ACS population, an up-to-date, large-scale randomized controlled clinical study with stratification of patients depending on risk and time from symptom onset would be required to evaluate the clinical outcomes of pretreatment in this clinical setting.

## Figures and Tables

**Figure 1 jcm-12-02374-f001:**
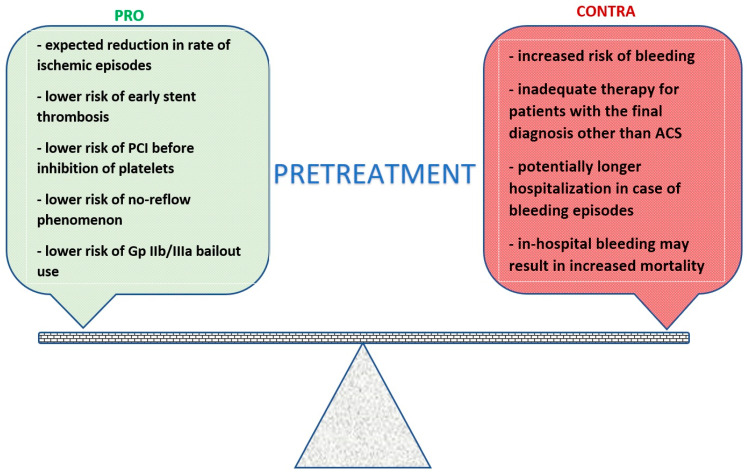
Commonly raised arguments for and against pretreatment in ACS patients. PCI—percutaneous coronary intervention, Gp—glycoprotein, ACS—acute coronary syndrome.

**Table 1 jcm-12-02374-t001:** A list of the most common clinical conditions to be considered in diagnostics procedures in suspected NSTE-ACS.

Differential Diagnoses of NSTE-ACS	
Cardiac	Non-Cardiac
○ Myocarditis ○ Pericarditis ○ Cardiac trauma ○ Takotsubo syndrome ○ Tachyarrhythmias ○ Aortic dissection ○ Acute heart failure ○ Coronary spasm ○ Cardiomyopathies	○ Pulmonary embolism ○ Tension pneumothorax ○ Pneumonia ○ Pleuritis ○ Gastrointestinal reflux ○ Peptic ulcer ○ Cholecystitis ○ Pancreatitis ○ Chest trauma ○ Musculoskeletal disorders

**Table 2 jcm-12-02374-t002:** A summary of clinical trials and meta-analyses presenting effects of pretreatment with P2Y_12_ receptor inhibitors.

	Author/Study	Agent Used in Pretreatment	Condition	Efficacy Outcome	Safety Outcome	Reference
1.	Uwe/CIPAMI	clopidogrel	STEMI	ns	ns	[26]
2.	Ducci/Load&Go	clopidogrel	STEMI	ns	ns	[29]
3.	Montalescot/ATLANTIC	ticagrelor	STEMI	ns	*p* = 0.008 at 24 h, *p* = 0.02 at 30 days	[30]
4.	Sabatine/PCI-CLARITY	clopidogrel	STEMI	*p* = 0.001 (less CV death, MI, or stroke before and after PCI)	ns	[30]
5.	Dörler/Austrian Registry	clopidogrel	STEMI	Lower rate of in-hospital mortality (*p* < 0.01)	Risk of bleeding not significantly increased (*p* = 0.90)	[27]
6.	Koul/SCAAR	clopidogrel	STEMI	1-year death/MI and 1-year death alone significantly reduced	Bleeding risk—ns	[28]
7.	Schupke/ISAR-REACT 5	ticagrelor	ACS	Pretreatment associated with worse outcomes; prasugrel (no-pretreatment) better than ticagrelor (pretreatment)—fewer deaths, MI or stroke; *p* = 0.006—see text	ns	[32]
8.	Montalescot/ACCOAST	prasugrel	NSTE-ACS	Early pretreatment—24% risk reduction (*p* = 0.004)	CABG-related and non-CABG-related TIMI major bleedings increased at day 7 (*p* = 0.006) and day 30 (*p* = 0.002) Early pretreatment—ns (*p* = 0.23)	[35]
9.	Dworeck/SCAAR	43.7% clopidogrel, 54.5% ticagrelor, 1.8% prasugrel	NSTE-ACS	ns	All bleedings: *p* = 0.02 only major bleeding: *p* = 0.002	[36]
10.	Steinhubl/CREDO	clopidogrel	NSTE-ACS	Fewer deaths, MI, TVR if administered >6 h prior to PCI (*p* = 0.051)	ns	[37]
11.	Tarantini/DUBIUS	ticagrelor, prasugrel	NSTE-ACS	ns	ns	[38]
12.	Bellemain-Appaix/meta-analysis	clopidogrel, prasugrel	ACS	No reduction in all-cause mortality (*p* = 0.24)	30–45% increased risk of bleeding (*p* < 0.0001)	[39]
13.	Bellemain-Appaix/meta-analysis	clopidogrel	ACS and CCS	Fewer CV episodes (*p* < 0.001), no reduction in deaths (*p* = 0.17)	ns	[40]
14.	Nairooz/meta-analysis	clopidogrel	ACS	Fewer MACEs: *p* < 0.0001 Lower mortality: *p* = 0.0003	ns	[41]

ACS—acute coronary syndrome, CV—cardiovascular, MACE—major adverse cardiovascular events, MI—myocardial infarction, ns—non-significant, NSTE-ACS—non-ST-elevation acute coronary syndrome, STEMI—ST segment elevation myocardial infarction, TVR—target vessel revascularization.

## Data Availability

No new data were created or analyzed in this study. Data sharing is not applicable to this article.

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
