# Peer review of "Pretreatment with P2Y12 Receptor Inhibitors in Acute Coronary Syndromes—Is the Current Standpoint of ESC Experts Sufficiently Supported?"

_jcm, 2023, doi:10.3390/jcm12062374_

Round 1
Reviewer 1 Report
It is a well written article and is relevant to the current clinical practice.
I have some minor suggestions
1. The authors should briefly discuss the rationale for pretreatment with a P2Y12 receptor blockers with the respect to achieving superior and early platelet inhibition. The authors should discuss early GPIIb/IIa trials followed by PCI-CURE trial that provided evidence for pretreatment with cloipdogrel. Since clopidogrel requires upto 24 hours with 300 mg LD or nearly 5-7 days with 75 mg MD, this trial demonstrated the use of pretreatment. However, loading dose of prasugrel or ticagrelor requires only 1-2 hours to achieve superior platelet inhibition, pretreatment may not be necessary. The authors should discuss this briefly in the introduction.
Moreover, the potential role of new P2Y12 receptor inhibitor, selatogrel. This agent is associated with fast and irreversible P2Y12 inhibition and may be a best option for pretreatment. This can be self administered as a subcutaneous injection.
Another concern is there are many mistakes in grammar that should be addressed. Please see attached file for further comments

Author Response
Dear Reviewer,
please find enclosed a document with replies to your review.
Best regards,
Piotr Niezgoda

Reviewer 2 Report
Pretreatment in ACS, especially in NSTEMI si always a question in ER and situantions when patient doesnt go to coronary angiography immediately.
Authors gave a good review of current research and guidelines. However, this is a complex area of investigation and summary, conclusion should be more thoroughly discussed (in view of this question that were raised by author themselves).
Author Response
Dear Reviewer,
Please find enclosed a document with the reply to your review.
Best regards,
Piotr Niezgoda
